# Semi-Physical Simulation of Fan Rotor Assembly Process Optimization for Unbalance Based on Reinforcement Learning

Huibin Zhang [1], Mingwei Wang [1,*], Zhiang Li [1], Jingtao Zhou [1], Kexin Zhang [1], Xin Ma [1] and Manxian Wang [2]

1    School of Mechanical Engineering, Northwestern Polytechnical University, Xi'an 710072, China;
     zhanghb1993@outlook.com (H.Z.); lza15735043977@163.com (Z.L.); zhoujt@nwpu.edu.cn (J.Z.);
     zkx2496228439@163.com (K.Z.); mx3013211568@163.com (X.M.)
2    AECC Xi'an Aero-Engine Ltd., Xi'an 710072, China; wmx18392975698@163.com
*    Correspondence: zhb1044439925@mail.nwpu.edu.cn

**Abstract:** An aero engine fan rotor is composed of a multi-stage disk and multi-stage blades. Excessive unbalance of the aero engine fan rotor after assembly is the main cause of aero engine vibration. In the rotor assembly process, blade sequencing optimization and multi-stage blade set assembly phase optimization are important for reducing the overall rotor unbalance. To address this problem, this paper proposes a semi-physical simulation method based on reinforcement learning to optimize the balance in the fan rotor assembly process. Firstly, based on the mass moments of individual blades, the diagonal mass moment difference is introduced as a constraint to build a single-stage blade sorting optimization model, and reinforcement learning is used to find the optimal sorting path so that the balance of the single-stage blade after sorting is optimal. Then, on the basis of the initial unbalance of the disk and the unbalance of the single-stage blade set, a multi-stage blade assembly phase optimization model is established, and reinforcement learning is used to find the optimal assembly phase so that the overall balance of the rotor is optimal. Finally, based on the collection of data during the assembly of the rotor, the least-squares method is used to fit and calculate the real-time assembly unbalance to achieve a semi-physical simulation of the optimization of balance during the assembly process. The feasibility and effectiveness of the proposed method are verified by experiments.

**Keywords:** aero engine rotor; blade sequencing optimization; reinforcement learning; unbalance; semi-physical simulation

## 1. Introduction

An aero engine is a device that provides power for an aircraft and has a decisive influence on the performance of the aircraft. One of the core problems currently limiting the development of the engine is the vibration of the rotor structure when rotating at high speed. According to the statistics, the number of engine failures caused by vibration accounts for 50% to 60% of the total number of engine failures [1]. It has been proven that rotor unbalance is the main factor that generates vibration during the operation of an aero engine [2].

An aero engine fan rotor is composed of a multi-stage disk and multi-stage blades [3]. Due to the manufacturing error of parts, the centroid of the disk and blade group deviates. If the assembly is done blindly, there is a high probability that the rotor unbalance will be excessive. To avoid the occurrence of the unbalance phenomenon, the most fundamental measure is to change the mass distribution of the rotor, eliminate the unbalance force, so that the rotor is balanced [4,5]. In the rotor assembly process, the assembly sequence of the blade affects the deviation of the mass center position of the blade group, and the assembly phase angle of the multi-stage blade and multi-stage disk affects the overall mass distribution of the rotor, both of which are important elements for controlling the

overall mass distribution of the rotor. Through blade sorting to make the overall mass center deviation of the blade set smaller and make the blade set mass center position a specific angle (assembly phase angle) relative to the disk mass center position during assembly, the mass center deviation of each rotor part can be made offset so as to reduce the overall rotor unbalance. Therefore, a suitable method for optimizing the balance of the fan rotor assembly process is of great importance for the improvement of the rotor assembly performance.

The current process for optimizing balance in aero engine assembly plants has the following main problems: (1) The blade sorting process only considers flushness and randomly selects pieces from the blades according to the blade code. Due to the blade manufacturing error with individual differences, randomly selected parts not only cause assembly quality dispersion, but poor connection performance consistency and often need to be repeatedly disassembled and adjusted to meet the assembly balance requirements, resulting in low assembly efficiency and high cost due to its "trial and error" nature. (2) The assembly process does not take into account the phase of blade assembly at all levels and the influence of the initial unbalance of the disk on the overall balance of the rotor, which can easily lead to the problem of large eccentricity of the overall mass of the rotor after assembly. (3) The main work step of rotor assembly is the assembly of the blade; because the assembly performance of the blade assembly process cannot be monitored, resulting in a lot of reliance on manual experience, it is difficult to regulate the rotor assembly balance in real time. Even if the installation is carried out in accordance with the pre-determined order of blade arrangement, the phenomenon of excess unbalance occurs, making the rotor performance less reliable and more volatile.

For the optimization of balance in the fan rotor assembly process, this paper proposes a semi-physical simulation method based on reinforcement learning. Firstly, based on the mass moments of individual blades, the diagonal mass moment difference is introduced as a constraint to build a single-stage blade sorting optimization model, and reinforcement learning is used to find the optimal sorting path so that the balance of the single-stage blade after sorting is optimal. Then, on the basis of the initial unbalance of the disk and the unbalance of the single-stage blade set, a multi-stage blade assembly phase optimization model is established, and reinforcement learning is used to find the optimal assembly phase so that the overall balance of the rotor is optimal. Finally, based on the collection of data during rotor assembly, the least-squares fitting method is used to calculate the real-time assembly unbalance, and the actual assembly data are combined with the theoretical data to realize the semi-physical simulation of the optimization of balance during the assembly process.

## 2. Literature Review

This section discusses the current status of domestic and international research on the following three aspects: rotor balancing methods, blade matching methods, and assembly phase optimization methods.

### 2.1. Status of Rotor Balance Research

In recent years, domestic and foreign scholars have obtained many results in the field of rotor balancing. Goodman [6] proposed the least-squares and weighted least-squares methods to minimize the sum of squares of the residual vibrations at each side vibration point measured on the calibration front by applying a set of correction masses to the calibration front. Bin Guangfu et al. [7] proposed the vibrational circular balance method, which can roughly determine the approximate distribution of the unbalance, substantially improving the balance accuracy. P. Gnieka [8] proposed a modal equilibrium method without trial weight, where a system of algebraic equations is derived by modal analysis based on the rotor's vibration pattern and generalized masses and then the optimal counterweight is solved, and the rotor is balanced by increasing the counterweight. Zhengshi Liu et al. [9] proposed the relative coefficient method on the basis of the influence coefficient method to

improve the balance efficiency of the rotor by directly measuring the relative coefficients with a two-channel dynamic signal analyzer.

Relevant domestic and foreign research has improved the balance of rotors to a certain extent, but the calculation method is complicated, and when the number of rotor stages is large, manufacturers face a large workload, it is impossible to monitor the balance of the rotor during the assembly process, and it is easy to have more than one installation and more than one trial.

### 2.2. Status of Research on Blade Selection Methods

In terms of blade matching methods, Jianmin Yue et al. [10] used a stepwise tuning simulation algorithm, which gives two models of mass and frequency as ranking schemes for optimization under the requirement of frequency. Tang Shaojun et al. [11] established the fitness function of the blade ranking problem and the evaluation method of its optimization by dividing the hub into quadrants according to the number of blades and optimizing the ranking of a given set of blades using a genetic algorithm under the constraints of the smallest possible mass difference between quadrants and the large frequency difference between blades, which improved the solution efficiency to some extent. By analyzing the disadvantages of the combinatorial optimization method and traditional genetic algorithm in solving the blade sorting problem, Zhao Desheng [12] transformed the multi-objective optimization problem into a single-objective optimization problem, used the penalty function method for the design of the fitness function, and established a blade sorting model based on the improved genetic algorithm, and its sorting results were accurate and converged quickly using the improved genetic algorithm. Edward A. Thompson et al. [13] used a polynomial time-simulated annealing algorithm to solve a large combinatorial optimization problem for blades. Wangbai Pan et al. [14] combined the hybrid finite-element model (MDFEM), Gaussian regression, and a genetic algorithm to propose a new method for the optimization of detuned impeller blades, which solved the problem of the large computational effort of the enumeration method, improved the optimization efficiency of the detuned impeller, and verified the effectiveness of the method with the example of a turbine.

Domestic and foreign scholars have made great research progress in blade selection methods. However, the blade matching problem is a combinatorial optimization problem, and the difficulty of solving the matching problem grows geometrically with the number of rotor stages and parts. So, it is difficult to find the optimal solution to the problem by traditional methods, and it is easy to fall into the local optimum.

### 2.3. Research Status of Assembly Phase Optimization Method

In terms of assembly phase optimization, Ding et al. [15] investigated a multi-stage, rotation-optimized assembly technique with a deviation transfer model for rotating component assembly and demonstrated the feasibility and applicability of applying the optimization objective function to the deviation transfer. Yang [16–20] proposed a direct assembly optimization technique to control the transmission accumulation of assembly deviations based on the error transmission model of rotor connection, used statistical analysis to derive the error probability distribution for different assembly qualities [21], and simulated the assembly example by Monte Carlo method. Mao-Guo Cao [22] used Powell's method to optimize the angular mounting position of rotor parts to reduce the unbalance forces and moments acting on the bearings due to the unbalance measure during the high-speed rotation of the engine. On this basis, Lixin Li et al. [23] further improved the optimization of the balance by optimizing the angular mounting positions of the disks at all levels through a genetic algorithm.

The above method can solve the rotor assembly phase optimization problem to a certain extent, but the influence of the initial unbalance of the disk on the assembly phase is not considered, and the optimization results have errors. Moreover, the traditional

optimization algorithm can only search for a certain extreme value point in the solution space, so it is easy to fall into local, optimal solutions.

## 3. Rotor Semi-Physical Simulation Implementation Framework

The fan rotor blade assembly process is shown in Figure 1. The fan rotor is a multistage disk structure consisting of components such as disks and blades at various levels. The fan rotor assembly process is as follows: firstly, multi-stage disk assembly; secondly, single-stage blade matching, determining the order of blade arrangement and single-stage blade assembly; finally, multi-stage blade combination assembly. The method proposed in this paper treats the multi-stage disk as an already-assembled part in the implementation and assumes that the balance of the multi-stage disk is already in compliance with the requirements. On this basis, the influence of blade sequencing and the multi-stage blade assembly phase angle on the overall rotor unbalance during blade assembly is investigated, and the real-time assembly performance is monitored during the assembly process to provide optimization guidance for the subsequent assembly process.

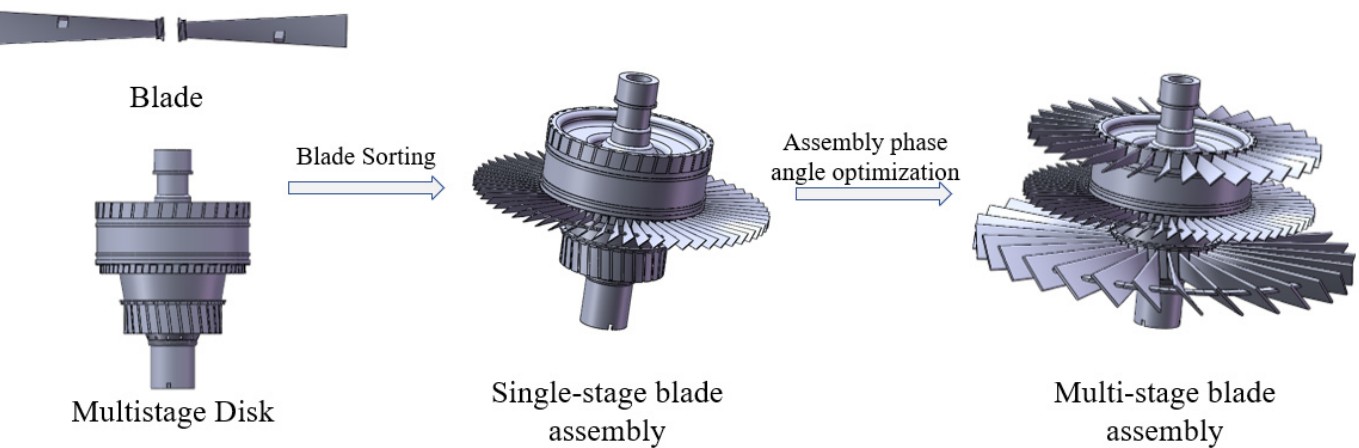

**Figure 1.** Fan rotor blade assembly process.

During the assembly process, the blades need to be assembled according to the predetermined blade layout sequence. The assembly result of each completed blade assembly is not only related to the current assembly process, but also to the assembly result of the previous step. Each blade affects the overall unbalance of rotor after assembly. In order to ensure that the final rotor balance meets the design requirements and to avoid excess assembly unbalance due to perceived installation errors and other factors, the assembly process needs to be monitored. Since the final overall unbalance is formed after the process is fully completed and the subsequent assembly steps are not performed during the assembly process, the actual measurement data of the subsequent steps cannot be obtained. Therefore, this paper constructs a semi-physical simulation model for single-stage blade assembly, combines the measured data of assembled blades with the theoretical data of unassembled blades, and calculates the overall unbalance of the rotor. The implementation process mainly includes:

(1) Establishing a single-stage blade sequencing optimization model. A mathematical model of single-stage blade sequencing is established with blade mass moment as input, diagonal position mass moment difference as a constraint, and minimum unbalance as the objective, and the blade sequencing order of each stage is optimized based on a reinforcement learning algorithm to reduce the unbalance of the blade set. The blade layout sequence is the main basis for blade assembly, while the blade set unbalance is also an input for the semi-physical simulation model;

(2) Calculation of the unbalance during assembly. As shown in Figure 2, when assembling to the $t$th blade, the previous $t-1$ work steps are performed, and the measured

data are used. The remaining work steps are replaced by theoretical scheduling data. The final unbalance value can be calculated by Equations (1) and (2).

$$\vec{U} = \vec{u} + \vec{u}'$$ (1)

$$\vec{u}' = \sqrt{\left(\sum_{i=t}^{n} m_i r_i \cos \alpha_i\right)^2 + \left(\sum_{i=t}^{n} m_i r_i \sin \alpha_i\right)^2}$$ (2)

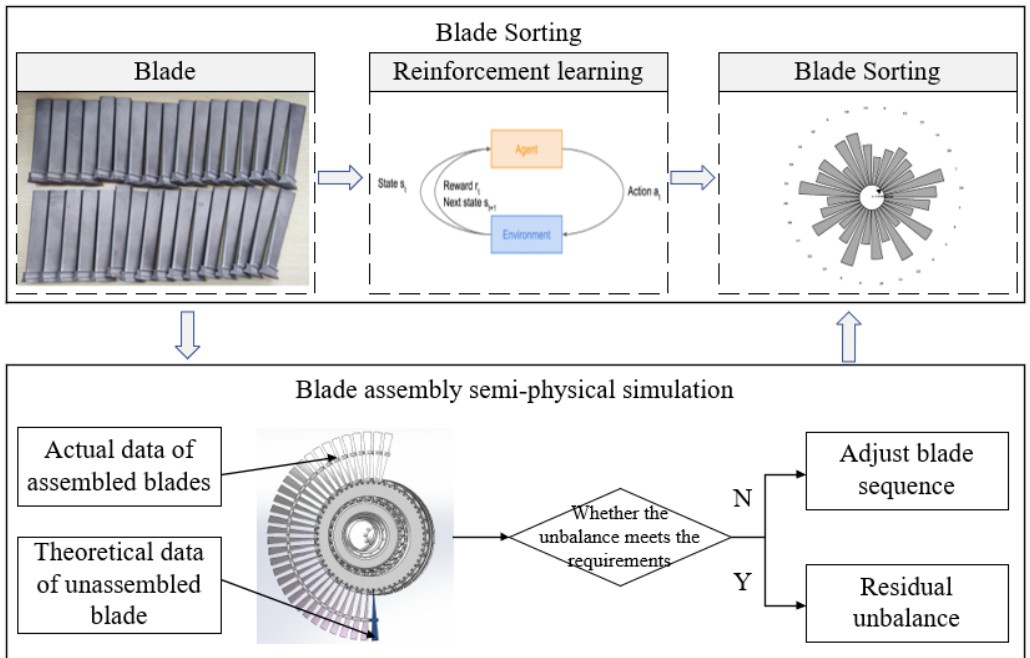

**Figure 2.** Single-stage blade semi-physical simulation.

In the formula, $\vec{U}$ is the remaining unbalance calculated by the blade assembly process; $\vec{u}$ is the measured value of the unbalance generated by the assembly steps; $\vec{u}'$ is the theoretical data of the unbalance of the later steps not performed;

(3) Optimization of unbalance. If the calculation result of unbalance during assembly exceeds the design requirement, the sequencing of the unassembled blades is re-executed, and the sequence of blades between the t + 1th work step and the nth work step is optimized until the design requirement is satisfied. If the calculation results of unbalance in the assembly process meet the design requirements, the assembly continues in the order of the original blades.

The above single-stage blade assembly is the assembly process because the fan rotor is usually composed of a multi-stage blade and multi-stage disk, so the multi-stage blade assembly process is completed before the fan rotor assembly, and each level of the blade assembly process is the same. In this paper, a multi-stage blade assembly phase optimization model is established to ensure that the overall balance of the final fan rotor meets the requirements. As shown in Figure 3.

The phase optimization of the multi-stage blade assembly is based on the analysis of the unbalance forces and torques generated by the residual unbalance of each stage's blade in space synthesis. Taking the residual unbalance of each blade as input and aiming to minimize the unbalance force and torque, a mathematical model of rotor multi-stage blade assembly phase optimization considering the initial unbalance of the disk is established.

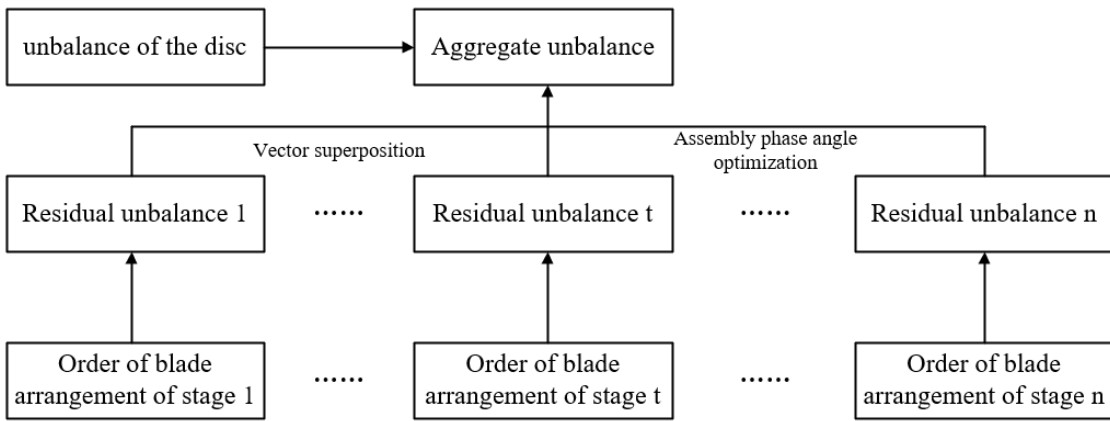

**Figure 3.** Fan rotor multi-stage blade assembly.

## 4. Single-Stage Blade Sequencing Optimization

The order in which the blades are arranged directly determines the size and direction of the unbalance and is one of the main factors affecting the balance of the fan rotor. The essence of blade sequencing optimization is to make the sum of the centrifugal forces of all blades on the axis of rotation as small as possible, thus, eliminating unbalanced forces and making the rotor's mass eccentricity smaller. In practice, due to the existence of machining errors, blade geometry, shape profile, and mass center distribution have small differences; it is difficult to achieve a rotor unbalance of 0 only through reasonable blade sorting optimization to make the rotor unbalance as small as possible.

### 4.1. Single-Stage Blade Sequencing Optimization Modeling

Based on the above analysis, the mechanical modeling of the blade sequencing problem is performed. When the fan rotor is rotating at angular speed $\omega$, the centrifugal force of each blade is expressed as:

$$\vec{f_i} = m_i \times \vec{r_i} \times \omega^2 \tag{3}$$

In the formula, $\vec{f_i}$ is the centrifugal force of the $i$th blade; $m_i$ is the mass of the $i$th blade; $\vec{r_i}$ is the vector diameter from the center of rotation of the $i$th blade to the center of mass; and $\omega$ is the angular velocity of rotor rotation.

The purpose of blade sequencing optimization is to eliminate the unbalanced force, that is, to minimize the centrifugal force, as shown in the following formula:

$$\min \sum_{i=1}^{n} \vec{f_i} = \min \sum_{i=1}^{n} m_i \times \vec{r_i} \times \omega^2 \tag{4}$$

Since the rotational angular velocity of all blades is the same, the minimum centrifugal force is actually to minimize the vector sum of the mass moments of all blades to the axis of rotation, so the objective function is:

$$\min \sum_{i=1}^{n} m_i \times \vec{r_i} \tag{5}$$

In the formula, $n$ is the number of blades; and $m_i \times \vec{r_i}$ is the mass moment of the $i$th blade, that is, the product of the distance between the blade centroid and the rotating center. From the derivation of Equations (4) and (5), the following can be seen: the process of single-stage blade sequencing actually balances the mass moment of each blade, and the mass moment vector sum of each blade is the remaining unbalance of the single-stage disk.

In order to make the rotor operation more stable, the deviation range of the blade set should be controlled to avoid the mass moment difference of the same group of blades

being too large. According to the enterprise design regulations, the blade diagonal position on the two blades of the mass moment difference cannot exceed a specific value. Therefore, the constraints of the blade sorting model in this paper are shown in the following equation:

$$\begin{aligned} \left| m_i r_i - m_j r_j \right| \leq \varepsilon \\ \left| i - j \right| = \frac{n}{2} \end{aligned} \tag{6}$$

In the formula, $\varepsilon$ is the upper limit of the mass moment difference on the diagonal position of the blade at a certain level.

### 4.2. Reinforcement Learning-Based Optimization Model Solving for Blade Sorting

In this section, a Q-learning learning algorithm is used to solve the blade sorting optimization model problem. Reinforcement learning addresses the entire blade sorting problem and is global in nature. The Q-learning learning algorithm can converge faster than ordinary reinforcement learning algorithms, which can improve the efficiency of blade matching.

#### 4.2.1. Reward and Punishment Function Design

The purpose of reinforcement learning in the blade ranking optimization model is to find a ranking strategy that minimizes the unbalance of blade groups. A good design of the reward and punishment function determines not only how well the model can be rewarded in the long run, but also how quickly the model can learn the ranking strategy. In order to express the reward and punishment function more clearly, the mass moment of the blade is decomposed orthogonally to the x-axis and y-axis, then, as shown in Figure 4, the sum of the x-axis and y-axis unbalance can be expressed as:

$$\begin{aligned} M_x = \sum_{i=1}^{n} m_i r_i \cos \alpha_i \\ M_y = \sum_{i=1}^{n} m_i r_i \sin \alpha_i \end{aligned} \tag{7}$$

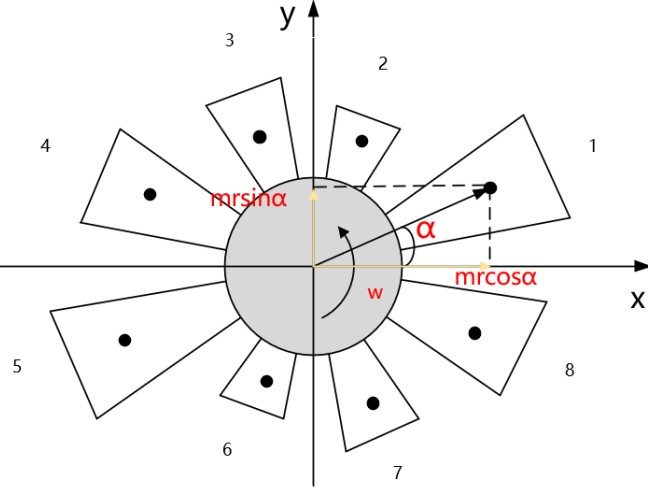

**Figure 4.** Blade mass moment orthogonal decomposition.

In the formula, $M_x$ denotes the sum of the mass moments in the x-axis direction; $M_y$ denotes the sum of the mass moments in the y-axis direction; $m_i r_i$ denotes the mass moment of the ith blade; and $\alpha_i$ denotes the angle between the center of mass of the ith blade and the positive direction of the x-axis.

In the Figure 4, the numbers 1–8 indicate 8 blades; $\alpha$ indicates the installation angle of the blades.

Thus, the equation for the unbalance of a single-stage blade can be expressed as:

$$M_{res} = \sqrt{M_x{}^2 + M_y{}^2} = \sqrt{\left(\sum_{i=1}^{n} m_i r_i \cos \alpha_i\right)^2 + \left(\sum_{i=1}^{n} m_i r_i \sin \alpha_i\right)^2} \qquad (8)$$

$$\beta = \arctan\left(\frac{M_y}{M_x}\right) \qquad (9)$$

In the formula, $M_{res}$ denotes the remaining unbalance after single-stage blade sorting; and $\beta$ denotes the position of the remaining unbalance.

The optimization objective considered in this paper is to minimize the unbalance value, so the reward and punishment function is set as:

$$R = -M_{res} = -\sqrt{\left(\sum_{i=1}^{n} m_i r_i \cos \alpha_i\right)^2 + \left(\sum_{i=1}^{n} m_i r_i \sin \alpha_i\right)^2} \qquad (10)$$

4.2.2. Q-Value Update Strategy

$R$ is the immediate reward for each action of the model and considers the long-term impact of each action. The matrix $Q_{h \times h}$ is the knowledge learned from experience, $h$ represents the number of leaves, the rows of the matrix $Q$ represent the current state, and the columns represent the possible actions to reach the next state; the matrix $Q$ is initialized to 0.

Every time the model is trained, the elements of the matrix $Q$ are updated by the following equation:

$$Q(s, a) = R(s, a) + \gamma \times \max(Q[next\ s, all\ a]) \qquad (11)$$

In the formula, $s$ is the current state; $a$ is the action; $R$ is the immediate reward; and $\gamma$ is the learning variable.

The value of the elements in the matrix $Q$ is then equal to the sum of the immediate reward $R$ under the current state s and action a and the learning variable $\gamma$ multiplied by the maximum reward value for all possible actions to reach the next state. The parameter $\gamma$ takes a value between 0 and 1, $0 \leq \gamma \leq 1$. If $\gamma$ is closer to 0, the model tends to consider only the immediate reward; if $\gamma$ is closer to 1, the model considers the overall reward with a greater weight.

The blade sorting optimization model learns from experience using the algorithm described above, with each experience equating to a training session. In each training session, the model continuously selects blades, superimposes moments of mass, and receives a reward value once it reaches the target state. The purpose of training is to enhance the experience of the model, represented by the matrix $Q$. More training results result in a more accurate matrix $Q$. If the matrix $Q$ is trained several times, then the model will not explore blindly during the blade sorting process but will reach the target state the fastest.

## 5. Multi-Stage Blade Assembly Phase Angle Optimization

*5.1. Multi-Stage Blade Assembly Phase Angle Optimization Modeling*

The blade sequencing optimization model established in the previous section can calculate the remaining unbalance after the blade arrangement of each level, while the overall unbalance produced by the aero engine fan rotor is the result of the accumulation of residual unbalance after the assembly of all blades. These residual unbalances produce large unbalance forces and unbalance moments when the rotor is rotating at high speed [24], and the engine performance is seriously affected under the unbalance forces and moments [22], as shown in Figure 5. Take the three-stage blade as an example; after each level of blade assembly, the blade group unbalance is superimposed on the xy plane. In the actual

assembly process, changing the blade assembly phase angle can effectively adjust the overall unbalance after multi-stage blade assembly, so it is necessary to consider the impact of blade assembly phase on the overall unbalance of the rotor.

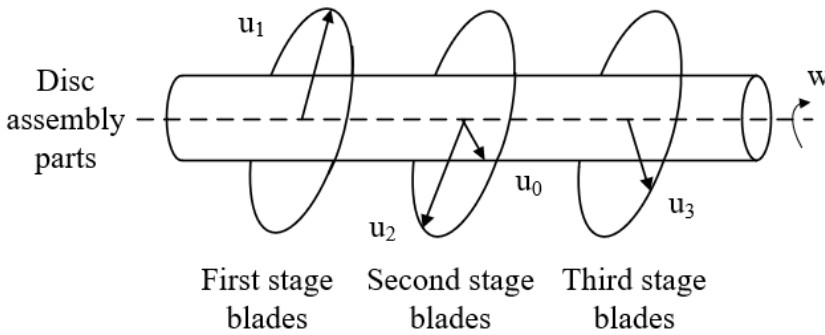

**Figure 5.** Superposition of unbalance schematic.

The size of the remaining unbalance after each stage of blade arrangement is $u_i(i = 1, 2, 3, \ldots, Z)$, and $u_0$ is the initial unbalance of the disk. Because of the unbalanced moment, a deformation force is generated at the journal position of the fan rotor so the position of the journal force in the rotor is recorded. Additionally, the distance coordinates of the other blade sets from the force point are defined as $l_i(i = 1, 2, 3, \ldots, Z)$. Taking the direction $\theta_i(i = 1, 2, 3, \ldots, Z)$ of each stage of blade imbalance as the design variable, the components of the magnitude of the unbalance force on the x- and y-axis are expressed as:

$$F_x = [u_0 + \sum_{i=1}^{Z} u_i \times \cos(\theta_i)] \times \omega^2 \tag{12}$$

$$F_y = \sum_{i=1}^{Z} u_i \times \sin(\theta_i) \times \omega^2 \tag{13}$$

In the formula, $F_x$ is the component of the unbalanced force on the x-axis; $F_y$ is the component of the unbalanced force on the y-axis; and $Z$ is the number of fan blade stages.

The moments of the unbalanced forces around the x-axis and y-axis are expressed as:

$$T_x = \sum_{i=1}^{Z} u_i \times \sin(\theta_i) \times \omega^2 \times l_i \tag{14}$$

$$T_y = [u_0 + \sum_{i=1}^{Z} u_i \times \cos(\theta_i)] \times \omega^2 \times l_i \tag{15}$$

In the formula, $T_x$ is the moment of the unbalanced force around the x-axis; and $T_y$ is the moment of the unbalanced force around the y-axis.

The purpose of multi-stage phase optimization is to minimize the amount of unbalance after multi-stage blade assembly, which means that the unbalance forces and moments are minimized. Therefore, the objective function of the multi-stage blade phase optimization model is:

$$\min F = \sqrt{F_x^2 + F_y^2} \tag{16}$$

$$\min T = \sqrt{T_x^2 + T_y^2} \tag{17}$$

*5.2. Multi-Stage Blade Assembly Phase Angle Optimization Model Solving*

The objective of reinforcement learning for multi-stage blade assembly phase angle optimization is to be able to find an assembly phase combination that minimizes the overall unbalance. Taking the three-stage blade as an example, this paper assumes that the reference plane where the initial unbalance force of the disk is located coincides with

the plane where the second-stage blade is located, and the distance between the first-stage blade and the third-stage blade and the reference plane is considered equal. A coordinate system is established on the datum, and the unbalance force $\vec{F_0}$ of the disk is set to coincide with the x-axis, then, the equation of the unbalance force in the x-axis and y-axis can be simplified as:

$$F_x = F_0 + \sum_{i=1}^{3} u_i \times \cos(\theta_i) \times \omega^2 \qquad (18)$$

$$F_y = \sum_{i=1}^{3} u_i \times \sin(\theta_i) \times \omega^2 \qquad (19)$$

The equation for the unbalanced moments around the x-axis and y-axis is simplified as:

$$T_x = [u_1 \times \sin(\theta_1) + u_3 \times \sin(\theta_3)] \times \omega^2 \times l \qquad (20)$$

$$T_y = [u_1 \times \cos(\theta_1) + u_3 \times \cos(\theta_3)] \times \omega^2 \times l \qquad (21)$$

The objective function of the multi-stage blade phase optimization model can be simplified as:

$$
\begin{aligned}
\min F &= \sqrt{F_x^2 + F_y^2} \\
&= \sqrt{\left(F_0 + \sum_{i=1}^{3} q_i \times \cos(\theta_i) \times \omega^2\right)^2 + \left(\sum_{i=1}^{3} q_i \times \sin(\theta_i) \times \omega^2\right)^2}
\end{aligned}
\qquad (22)
$$

$$
\begin{aligned}
\min T &= \sqrt{T_x^2 + T_y^2} \\
&= \omega^2 \times l \times \sqrt{(q_1 \times \sin\theta_1 + q_3 \times \sin\theta_3)^2 + (q_1 \times \cos\theta_1 + q_3 \times \cos\theta_3)^2}
\end{aligned}
\qquad (23)
$$

The optimization objective considered in this paper is to minimize the amount of unbalance after multi-stage blade assembly, that is, to minimize the unbalance forces and moments. Therefore, the Q-learning reward and punishment function is set as follows:

$$R = -(\eta_1 F + \eta_2 T) \qquad (24)$$

In the formula, $\eta_1, \eta_2$ are the weights of the reward factors.

The Q-value update strategy for the multi-stage blade phase optimization model is the same as described in Section 4.2.2. By training the matrix $Q$ several times, the model reaches the optimal state quickly in the multi-stage blade phase optimization process.

## 6. Instance Validation

### 6.1. Experimental Preparation

This paper takes the assembly of a certain stage of an aero engine fan rotor blade as an example verification object. The experimental platform is a common lathe (CS6150), and the rotor material is aluminum alloy (AlSi10Mg). A laser displacement sensor (HG-C1030) is used to measure the displacement of rotor vibration. The sensor is fixed directly to the fixture, and the data collected by the sensor are stored by the microcontroller. The microcontroller is connected to the PC via the USB interface, and the data are displayed on the XCOM software in the PC. The rotor speed is set to 100 r/min, and the sampling frequency of the sensor is 800 HZ. The specific installation of the experimental device is shown in Figure 6.

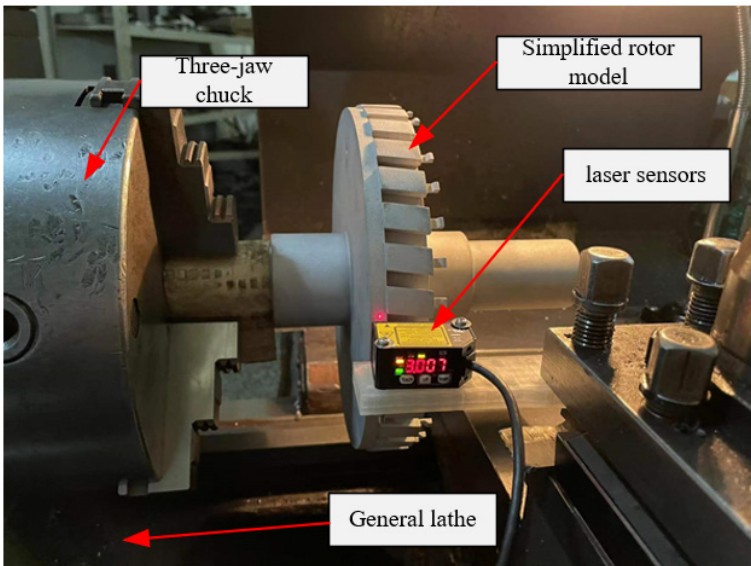

**Figure 6.** Experimental setup diagram.

Before the experiment starts, the mass moment data of the blade are measured, and the data are obtained by using the mass moment measuring instrument (MW0) from Schenker, Germany, and the blade data are shown in Table 1. The blades are numbered according to the mass moment data from smallest to largest, as shown in Figure 7.

**Table 1.** Blade data sheet.

| Blade Number | Mass Moment (g·mm) | Blade Number | Mass Moment (g·mm) |
|---|---|---|---|
| 1 | 5238 | 18 | 5253 |
| 2 | 5177 | 19 | 5253 |
| 3 | 5146 | 20 | 5177 |
| 4 | 5253 | 21 | 5238 |
| 5 | 5268 | 22 | 5192 |
| 6 | 5146 | 23 | 5268 |
| 7 | 5268 | 24 | 5268 |
| 8 | 5268 | 25 | 5101 |
| 9 | 5283 | 26 | 5131 |
| 10 | 5192 | 27 | 5238 |
| 11 | 5283 | 28 | 5238 |
| 12 | 5162 | 29 | 5192 |
| 13 | 5238 | 30 | 5192 |
| 14 | 5223 | 31 | 5207 |
| 15 | 5283 | 32 | 5177 |
| 16 | 5268 | 33 | 5192 |
| 17 | 5253 | | |

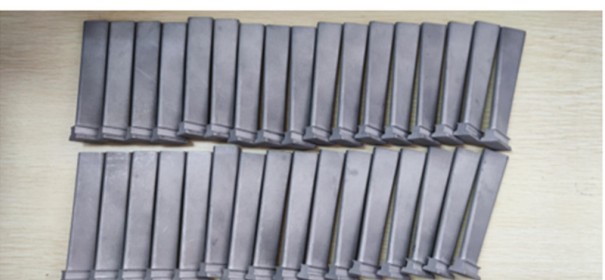
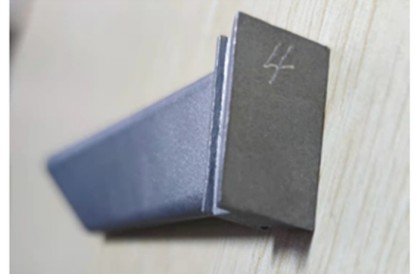

**Figure 7.** Schematic diagram of the blade.

### 6.2. Single-Stage Blade Sequencing Optimization

Experimental validation of the blade sorting optimization model after the blade data and the initial unbalance of the disk are known. The reinforcement learning algorithm is used to obtain the blade ranking order, install the blades on the disk in order, and compare them with the experimental measurements to verify the effectiveness of the blade ranking method.

The order of blade arrangement is shown in Figure 8, and the black triangle indicates the angle where the remaining unbalance is measured. The numbers in the figure indicate the angular value of the location of the black triangle.

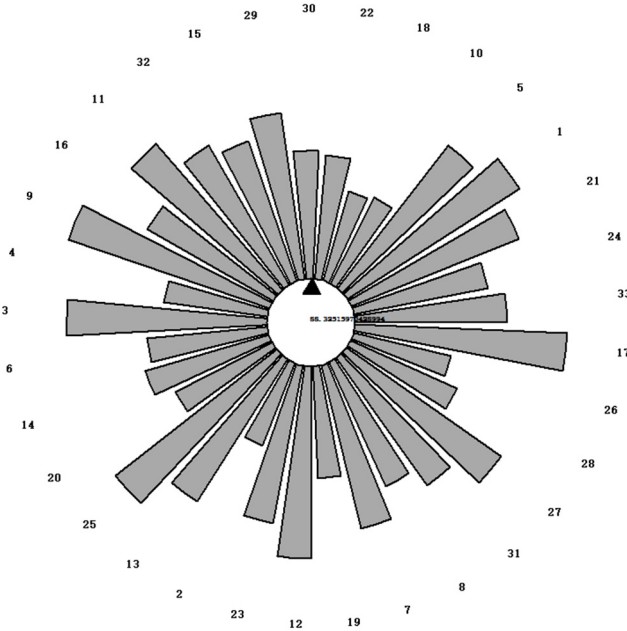

**Figure 8.** Blade layout diagram.

In order to characterize the blade optimization order more intuitively, the ranking order is presented as a numerical sequence, and the ranking order is 30-29-15-32-11-16-9-4-3-6-14-20-25-13-2-23-12-19-7-8-31-27-28-26-17-33-24-21-1-5-10-18-22. In order to verify the optimization effect of reinforcement learning, the results of the remaining unbalance and the diagonal position mass moment differences are obtained by ranking the blades according to several methods mentioned in the literature [21,22], as represented in Table 2.

**Table 2.** Comparison of results of sorting methods.

| Sorting Method | Weight Sorted [25] | Work-Shop [25] | Sorting-1 4skip [25] | Sorting-2 7skip [25] |
|---|---|---|---|---|
| Residual unbalance | 10,563.27 | 2162.00 | 3452.83 | 2241.50 |
| Mass moment difference | 1820 | 1200 | 2060 | 580 |
| Sorting Method | Sorting-3 [14] | Sorting-4 [14] | Sorting-5 [14] | Q-learning |
| Residual unbalance | 1126.86 | 947.02 | 807.56 | 4.95 |
| Mass moment difference | 340 | 580 | 340 | 720 |

As can be seen from Table 2, for Weight Sorted [25] and Sorting-1 (4skip) [25] in the blade arrangement process, the remaining unbalance value seriously exceeds the requirements, and the diagonal mass moment difference does not meet the requirements. Work-Shop [25], Sorting-2 (7skip) [25], Sorting-3 [14], Sorting-4 [14], and Sorting-5 [14] have a relatively low remaining unbalance under the condition where the diagonal mass moment

difference meets the requirement, but the results are still less than satisfactory. Then, a comparison is made between the Q-learning algorithm and the traditional sorting method. As can be seen from the table, the diagonal mass moment differences of the reinforcement learning meet the requirements, and the calculated values of the remaining unbalance are much smaller than those of the traditional ranking methods, thus proving the effectiveness of the proposed method in this paper.

### 6.3. Single-Stage Blade Sequencing Semi-Physical Simulation

To verify the validity of the semi-physical simulation model according to the blade installation sequence 30-29-15-32-11-16-9-4-3-6-14-20-25-13-2-23-12-19-7-8-31-27-28-26-17-33-24-21-1-5-10-18-22, 14 blades are assembled in advance, and, when blade numbered 2 is installed, it is wrongly installed as blade number 7, as shown in Figure 9.

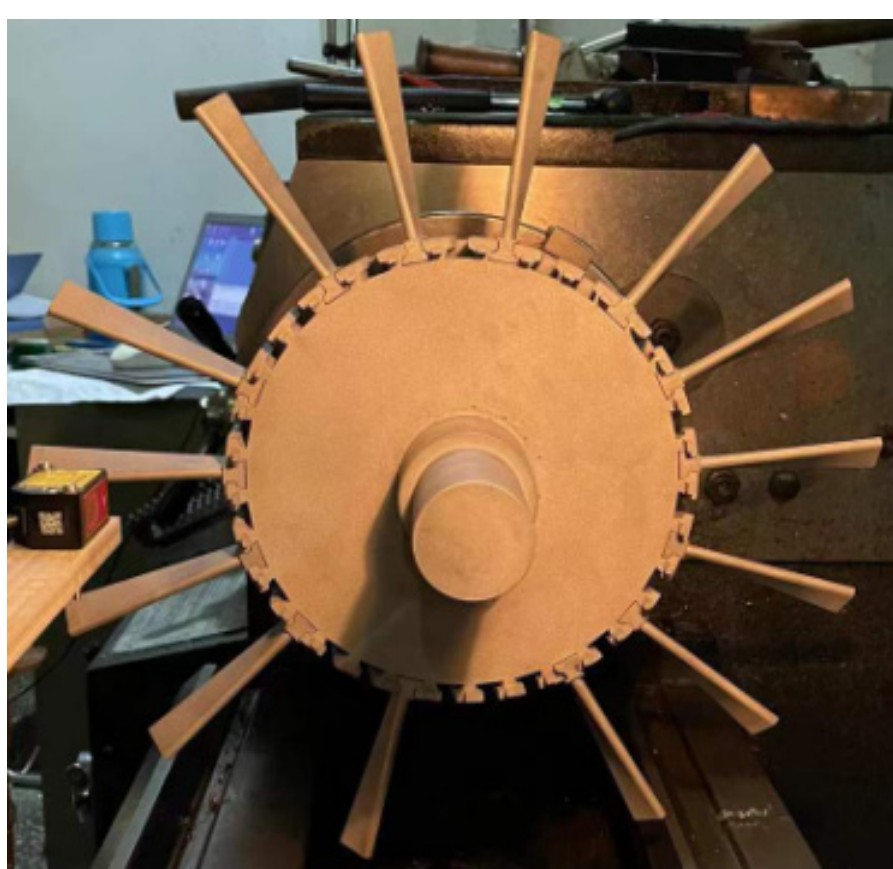

**Figure 9.** Blade assembly drawing.

Before the blades are assembled, the initial unbalance of the disk is calculated. The initial unbalance cannot be measured directly, so this paper calculates the initial unbalance indirectly by measuring the displacement change of the disk, which is measured as follows:

(1) The change in the radial displacement of the disk during rotation of the rotor is measured by a laser displacement sensor, and the value is recorded;

(2) The actual center of the mass position of the rotor is fitted using the least-squares method [16] to derive the eccentricity of the disk, as shown in Figure 10. Black point indicate theoretical rotor center of mass position, red point indicate actual rotor center of mass position;

(3) After the eccentricity distance is known, the initial unbalance of the disk is calculated by Equation (25), and the data are shown in Table 3.

$$M' = m'e' \tag{25}$$

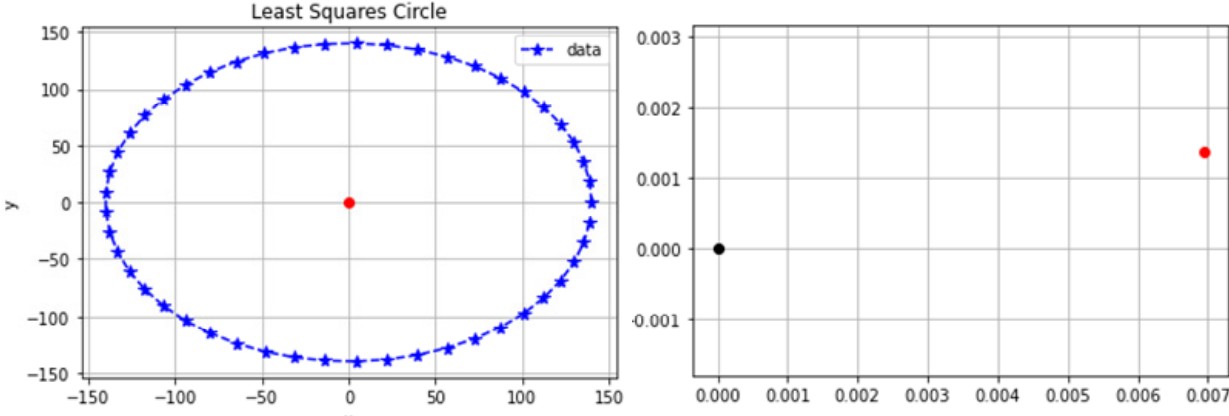

**Figure 10.** Least-squares circle fitting.

**Table 3.** Data of the disk.

| Weight of the Disk (g) | Eccentricity (mm) | Initial Unbalance (g·mm) |
| :---: | :---: | :---: |
| 5035.2 | 0.005422 | 27.30 |

In the formula, $M'$ is the initial unbalance of the disk; $m'$ is the mass of the disk; and $e'$ is the eccentricity of the disk, that is, the distance between the center of mass of the disk and the axis of rotation.

When the blade numbered 16 is wrongly installed as blade number 13, the overall remaining unbalance is calculated by the semi-physical simulation model to be 549.2, which exceeds the value of permissible unbalance. To avoid repeated disassembly of the blade, the overall unbalance is adjusted by optimizing the order of the unassembled blade, and the blade order is adjusted before and after, as shown in Table 4.

**Table 4.** Order of blade arrangement.

| Adjust the Blade | Blade Order |
| :---: | :---: |
| Before adjustment | 30-29-15-32-11-16-9-4-3-6-14-20-25-13-2-23-12-19-7 -8-31-27-28-26-17-33-24-21-1-5-10-18-22 |
| After adjustment | 30-29-15-32-11-16-9-4-3-6-14-20-25-13-7-22-5-2-19 -8-24-31-33-23-28-18-27-17-21-1-12-10-26 |

After calculation, the unbalance of the blade assembly after the wrong adjustment sequence is 17.71, which meets the requirement of balance, indicating that the constructed semi-physical simulation model can realize the monitoring of the unbalance in the blade assembly process.

### 6.4. Multi-Stage Blade Assembly Phase Angle Optimization

Taking the first-, second-, and third-stage blade data of the same rotor as examples, the optimized blade layout diagram derived by Q-learning reinforcement learning is shown in Figure 11. After the completion of the blade arrangement at all levels, the assembly phase angle between disk and blades at all levels is optimized through reinforcement learning. The initial unbalance force of the disk and the remaining unbalance force vector of the blades at each level are superimposed, which finally makes the remaining unbalance force and moment of the whole rotor meet the requirements.

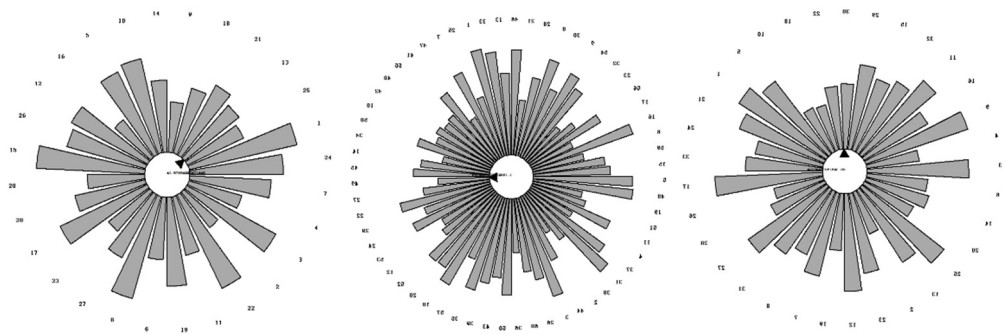

**Figure 11.** The first-, second-, and third-level blade arrangement diagrams.

The sequence table (counterclockwise) of the first-, second-, and third-level blades optimized using reinforcement learning is shown in Table 5.

**Table 5.** First-, second-, and third-level blade arrangement order table.

| Number of Blade Stages | Arrangement Order |
|---|---|
| First-stage blades | 17-3-11-28-27-6-7-16-13-1-19-5-2-12-24-8-9-15-25-23-14-18-10-26-4-22-20-21 |
| Second-stage blades | 55-46-31-51-23-60-40-54-9-56-47-39-10-21-13-48-43-32-20-27-41-26-18-33-12-11-8-19-34-5-15-36-38-4-3-57-24-7-14-29-50-22-6-28-17-49-52-35-45-16-25-59-42-30-37-1-44-2-58-53 |
| Third-stage blades | 9-10-5-27-33-14-30-18-15-29-26-11-23-8-3-19-20-24-28-1-13-4-21-7-25-22-2-31-12-32-6-16 |

By optimizing the multi-stage blade assembly phase angle through reinforcement learning, the overall residual unbalance force and unbalance moment and the blade assembly phase angle of each stage before and after optimization are compared, as shown in Table 6.

**Table 6.** Comparison of results before and after optimization.

| | | Before Optimization | After Optimization |
|---|---|---|---|
| Input | Disk initial unbalance force | 0.8788 | 0.8788 |
| | Remaining unbalance force of the first-stage blade | 0.0450 | 0.0450 |
| | Remaining unbalance force of the second-stage blade | 0.0892 | 0.0892 |
| | Remaining unbalance force of the third-stage blade | 0.0293 | 0.0293 |
| Output | Overall residual unbalance force | 1.3633 | 0.7008 |
| | Overall residual unbalance moment | 1.2523 | 0.0046 |
| | First, second, third installation phase angle | 0/0/0 | 328/113/180 |

Analyzing the data in Table 6, before optimization, the maximum remaining unbalance force calculated with the phase angle of the first-, second-, and third-stage blade set installation is 1.3633 at 0°, the unbalance moment is 1.2523, and the remaining unbalance is 150.12, which obviously exceeds the design value of 100 for the overall remaining unbalance of an aero engine plant. The optimized installation angle of the first-stage blade set relative to the disk is 328°, the angle of the second-stage blade set relative to the disk installation is 113°, and the angle of the third-stage blade set relative to the installation disk is 180°, and the remaining unbalance is 77.17, which meets the overall remaining unbalance requirement. As shown above, it is feasible to optimize the phase angle of multi-stage blade assembly by using reinforcement learning.

## 7. Conclusions

In this paper, we proposed a semi-physical simulation method based on reinforcement learning to optimize balance in the fan rotor assembly process. Firstly, based on the mass moments of individual blades, the diagonal mass moment difference was introduced as a constraint to build a single-stage blade sorting optimization model, and reinforcement learning was used to find the optimal sorting path so that the balance of the single-stage blade after sorting is optimal. Then, on the basis of the initial unbalance of the disk and the unbalance of the single-stage blade set, a multi-stage blade assembly phase optimization model was established, and reinforcement learning was used to find the optimal assembly phase so that the overall balance of the rotor is optimal. Finally, based on the collection of data during the assembly of the rotor, the least-squares method was used to fit and calculate the real-time assembly unbalance to achieve a semi-physical simulation of the optimization of balance during the assembly process. The validity of the proposed method and model was verified through experiments. However, because the model used in the experiment was a simplified model, which is different from a real fan rotor, the model needs to be optimized in the future to adapt to the on-site fan rotor assembly of enterprises.

**Author Contributions:** H.Z. made contributions to the analysis of blade sorting problems, the determination of solutions to blade sorting problems, experimental design, experimental data processing, and paper writing. M.W. (Mingwei Wang) and J.Z. gave guidance on the overall methodology of the paper; Z.L., X.M. and K.Z. contributed to the figures, tables, and English translation of the paper; M.W. (Manxian Wang) provided the 3D model of the rotor. All authors have read and agreed to the published version of the manuscript.

**Funding:** This research was funded by the National Key R&D Program of China (no. 2019YFB1703802).

**Data Availability Statement:** The raw/processed data needed to reproduce these findings cannot be shared at this time because they are confidential.

**Acknowledgments:** Thanks to Enming Li and Changsen Yang for compiling the experimental data.

**Conflicts of Interest:** The authors declare no conflict of interest.

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
