# Peer review of "Semi-Physical Simulation of Fan Rotor Assembly Process Optimization for Unbalance Based on Reinforcement Learning"

_aerospace, doi:10.3390/aerospace9070342_

Round 1

Reviewer 1 Report

This study proposed a semi-physical simulation method based on reinforcement learning to optimize the unevenness of the fan rotor assembly process and validated with experiment. The topic of the paper is interesting and the manuscript is comprehensive and the results seem coherent and described with sufficient clarity. The conclusions of the paper are consistent, with generic value, being useful for engineers in the field. I only have three minor comments that can help to improve the manuscript:

1. Please carefully check the typos error, such as Eqs 2, page 12. There is no line number, so it is hard to mention the exact locations.

2. The quality of all figures is low. Please provide higher-quality figures.

3. Caption of tables, and figures should be checked carefully.

4. On page 13, in the first paragraph, the authors may explain Table 2 but the authors link it to Table 3. Please clarify. 

5. Many typo errors can be found on page 14.

Reviewer 2 Report

The optimal arrangement of blades in a multi-bladed rotor is an important and interesting problem that has received insufficient attention in the literature.  I therefore welcome this contribution which is substantially strengthened by the experimental work described in the second half of the paper and the consideration of multiple rotors on the same shaft.  I have, however, three main criticisms that must be addressed before the manuscript can be accepted for publication.

First, the manuscript in general, and the Introduction in particular, is vague and unspecific.  “unevenness” is never defined and statements like “Each blade is assembled to produce a corresponding unevenness” on page 4 do not help the reader to understand the problem.  As far as I can tell there are three main components to the problem:

1. The number of blades is large enough that analytic methods of minimizing imbalance are not applicable.  These are described in the wind turbine context by Hitz & Wood (2010).

2. The blades are too expensive to be thrown away if they exceed some threshold of imbalance.

3. The assembly of the blades is arbitrary.  In other words, any blade can be attached anywhere on the rotor.

Second, the statement of the problem at the start of section 4.1 must be improved.  It seems to me an equally useful goal would be to minimize the eccentricity of the rotor centre of mass and it is not necessary to balance opposing blades, Hitz & Wood (2011).  The authors may have good reasons to use their methodology but these need to be explained carefully.

Thirdly, and following the comments made above, there are lapses in the English expression and the equations are poorly formatted with undefined variables and significant repetition.  Some examples of poor expression are the use of author first names in referencing, “et al” for “et al.”, the indenting of “where” and sometimes “Where” below equations, and statements like “the higher the accuracy of the rotor” on page 6.  Again “accuracy” is not defined or made specific. The Author Contributions mention a thesis without any explanation.

Round 2

Reviewer 2 Report

The authors' response to my comments was detailed and appropriate.  The manuscript can be accepted for publication.